# High-speed 4D neutron computed tomography for quantifying water dynamics in polymer electrolyte fuel cells

Ralf F. Ziesche [1,2,3,4], Jennifer Hack[1], Lara Rasha [1], Maximilian Maier[1], Chun Tan [1,2], Thomas M. M. Heenan [1,2], Henning Markötter [5,6,7], Nikolay Kardjilov [5], Ingo Manke [5], Winfried Kockelmann[4], Dan J. L. Brett [1,2] & Paul R. Shearing [1,2✉]

In recent years, low-temperature polymer electrolyte fuel cells have become an increasingly important pillar in a zero-carbon strategy for curbing climate change, with their potential to power multiscale stationary and mobile applications. The performance improvement is a particular focus of research and engineering roadmaps, with water management being one of the major areas of interest for development. Appropriate characterisation tools for mapping the evolution, motion and removal of water are of high importance to tackle shortcomings. This article demonstrates the development of a 4D high-speed neutron imaging technique, which enables a quantitative analysis of the local water evolution. 4D visualisation allows the time-resolved studies of droplet formation in the flow fields and water quantification in various cell parts. Performance parameters for water management are identified that offer a method of cell classification, which will, in turn, support computer modelling and the engineering of next-generation flow field designs.

[1] Electrochemical Innovation Lab, Department of Chemical Engineering, UCL, London WC1E 7JE, UK. [2] The Faraday Institution, Quad One, Harwell Science and Innovation Campus, Didcot OX11 0RA, UK. [3] Diamond Light Source Ltd, Harwell Science and Innovation Campus, Didcot, OX 11 0DE, UK. [4] STFC, Rutherford Appleton Laboratory, ISIS Facility, Harwell OX11 0QX, UK. [5] Helmholtz-Zentrum Berlin für Materialien und Energie (HZB), Hahn-Meitner-Platz 1, 14109 Berlin, Germany. [6] Technische Universität Berlin, Straße des 17. Juni 135, 10623 Berlin, Germany. [7] Bundesanstalt für Materialforschung und -Prüfung, Unter den Eichen 87, 12205 Berlin, Germany. ✉email: p.shearing@ucl.ac.uk

Fuel cells are electrochemical energy conversion devices that can offer a reliable low-carbon alternative for continuous electrical power output, provided that substantial performance drops due to fuel starvation, catalyst degradation or water saturation are avoided. There are several types of fuel cells that are being deployed for commercial applications, among which the polymer electrolyte fuel cell (PEFC) is one of the most promising, due to a relatively low operating temperature (60–100 °C) and ease of fabrication.

PEFCs typically employ a per-fluorinated sulfonic acid membrane that is sandwiched by two platinum-containing catalysts layers (CLs), which constitute the anode and cathode. Carbon-based gas diffusion layers (GDLs) are used as an intermediate layer between the CL and flow fields, which act as both current collector as well as providing a pathway for gas supply to the CLs and water removal into the flow fields. The two electrodes, GDLs and membrane together constitute the membrane electrode assembly (MEA). Hydrogen is supplied to the anode where the oxidation reaction splits the hydrogen into protons and electrons. The protons are conducted through the membrane to the cathode where the reduction reaction with oxygen forms water. In order for the protons to be transported, the membrane must be sufficiently hydrated, and therefore water can be found on both the anode and cathode during operation.

The flow fields are considered the 'lungs' of the fuel cell, since they deliver gas to the GDLs, act as current collectors and, crucially, provide a route for water removal out of the cell. Various flow field designs have been explored in order to optimise these delivery/collection transport processes[1,2]. However, knowledge of the exact whereabouts of water accumulation dynamics during cell operation remains scarce. Optimising these various transport properties can ultimately produce cells with higher power densities and thus greater competitiveness within the mass market.

Computed tomography (CT) has emerged as an essential tool for the three-dimensional characterisation of electrochemical device microstructures[3]. Furthermore, various probing methods are employed, for instance, X-ray beams produce tomograms by interacting with the atomic electrons of material, whereas neutron beams produce analogous 3D views via interactions with the sample's atomic nuclei.

X-ray CT has been widely used to study various operational and degradation processes occurring within PEFCs using in situ or *operando* methods. These require bespoke cells, optimised for imaging conditions by using small sizes to fit into the lab- or synchrotron-based imaging cavity, whilst still being representative of larger systems[4–10]. A significant focus of this in situ/ *operando* work has been to study the accumulation of water within the MEA. Given the available resolution of X-ray CT, a typical tomography scan will focus on microscale features, like the fibres of the GDL, the CL as a bulk layer, but with only one or two flow channels fitting within the field of view (FoV). Thus, work has mainly focussed on imaging the evolution of water within the GDL, with studies demonstrating the preference for formation under the land regions of the flow field[6,11,12] or highlighting the positive effect of controlling pore sizes within the MEA to aid water removal[13]. However, there are limitations of imaging water transport with X-rays due to the poor contrast between water and carbon fibres of the GDL, a limited FoV that is a trade-off between high spatial and temporal resolution and radiation induced material damage by high intensity synchrotron X-ray beams, often leading to a drop in cell performance[14,15].

Consequently, in order to study water formation across the entire flow field area, neutron imaging is increasingly used; neutrons interact strongly with the hydrogen atoms in water providing good contrast between water and the PEFC. Furthermore, neutron imaging provides a larger FoV, albeit at a loss of spatial resolution, such that information about the entire flow field/MEA area is obtained. The vast majority of studies employing neutrons for imaging PEFCs have used radiography, in either through- or in-plane mode. Consequently, the resulting image greyscale values arise from path-integrals through the plane of investigation, and thus the volume of water residing in the MEA and anode/cathode flow fields cannot be easily distinguished. Through-plane imaging studies are commonly used to study the transport of water along flow channels[16,17] and the water thickness through the cell can be calculated[18,19]. However, as mentioned, differentiation between the cathode and anode flow channels is not possible and practitioners tend to adopt techniques like altering the anode flow field design[18] or changing its orientation[17], which reduces the representability of the cell. In-plane imaging can also be used to overcome the issue of distinguishing the anode from the cathode[16,20], but again this provides no information about droplet shape, volume or movement through the flow channels during operation.

Although there is a clear need to move towards neutron CT for imaging water evolution in PEFCs, 4D (i.e., three spatial dimensions plus time) neutron CT demonstrations are rare and limited by poor temporal resolution due to long acquisition times. A handful of studies have produced tomograms of PEFCs using neutron CT[21–27] but all of these were obtained ex situ, without temporal resolution and with long scan times over several hours. Furthermore, with the exception of the recent work by Alrwashdeh et al.[27], those studies go back nearly a decade or more.

In this work, we demonstrate the use of high-speed 4D neutron CT to map and quantify the water distribution within an operational PEFC, with high temporal and spatial resolution. By optimising the image quality using a high-flux neutron source and a bespoke mini-cell, water droplets could be resolved with respect to their position within the cell e.g. the anode and cathode flow field plates and the MEA. Through this characterisation methodology, we were able to quantify the correlations between operational parameters such as current and potential with respect to transient water dynamics. This work pioneers the use of neutrons for the 4D mapping of water within PEFCs. The knowledge gained through these experiments, and future ones based upon this methodology, will expand our understanding of transient water dynamics in PEFCs during operation. Such information will allow for improved cell designs, greater power densities, and extended lifetimes; all of which are essential for improving mass-market competitiveness.

## Results

For the 4D neutron imaging experiment, a miniaturised PEFC was designed and manufactured, based on previous designs for X-ray CT[7]. Figure 1a shows the shape and size of the gold-coated aluminium endplate design with a single serpentine flow field design. The cell uses an MEA with an active area of 2 cm[2] (see "Methods" section). In order to improve neutron transmission for optimum imaging quality, the through-plane cell thickness was reduced in the active area from ca. 14.4–8.8 mm. Figure 1b displays a 3D schematic of the fuel cell and its mounting onto the tomography rotation stage.

The high-speed 4D neutron CT was performed on the CONRAD-2 (V7) neutron imaging beamline at the BER II research reactor at the Helmholtz-Zentrum Berlin (HZB, Germany). The limited lengths of the connected pipes and cables necessitated rotation of the cell forwards and backwards by ±370° for tomography (see Fig. 1c). A more detailed overview of the CONRAD-2 beamline can be found elsewhere[28,29].

After cell conditioning and before imaging, polarisation curves were collected to understand the performance of the cell and

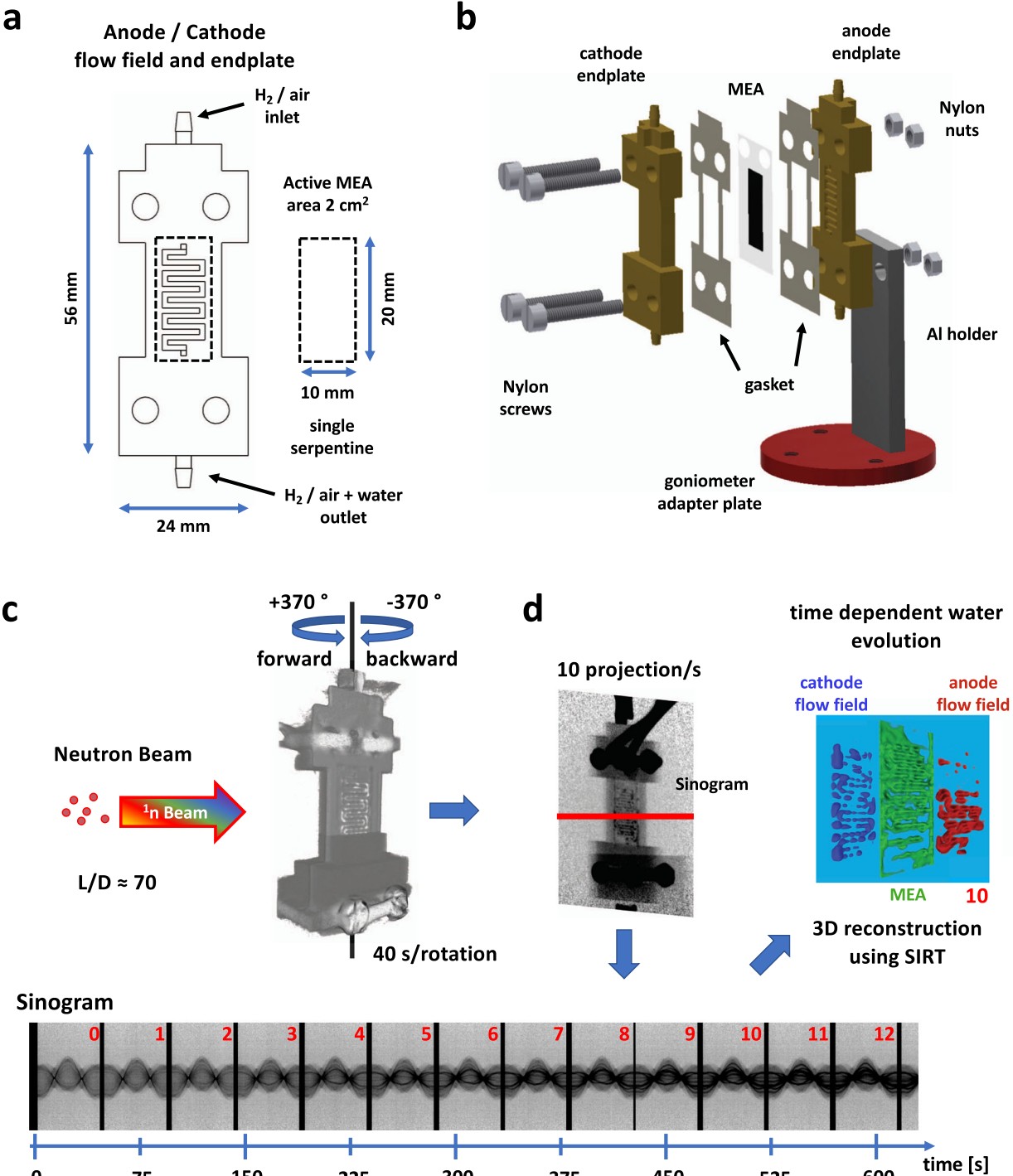

**Fig. 1 Single serpentine flow field design, fuel cell and imaging processing. a** The flow field design incorporated in the cell endplates for both anode and cathode with hydrogen and air in- and outlets, respectively. **b** Cell assembly on the rotation stage using nylon screws and nuts. **c** Tomograms were collected in 40 s during each of ±370° forward and backward rotations in the neutron beam with an L/D collimation of ca. 70. **d** Data processing, including flat fielding projections, generation of time-dependent sinograms (below) with water volume build-up (increasingly dark grey values from 0 to 12), 3D reconstruction, followed by water quantification in the anode and cathode flow fields, and in the MEA.

determine the cut-off current density at 0.3 V (see Supplementary Figs. 1 and 2). A range of galvanostatic and potentiostatic hold experiments were carried out, whilst constantly collecting 40-s tomograms. Constant currents were held for about 600 s between 100 and 700 mA cm$^{-2}$ in 100 mA cm$^{-2}$ steps and constant potentials were held for about 600 s at potentials of 0.7, 0.5 and 0.3 V. Figure 1c, d illustrates the construction of sinograms and 3D volumes for a sequence of 40 s tomograms.

Figure 2a, b shows the corresponding potential and current graphs of the potentiostatic and galvanostatic measurements. Potential and current over-shoots are clearly visible at the beginning of the cell operation and correspond to a hydrogen oversupply at the anode side, which stems from the hydrogen gas build-up before the cell operation started. The excess hydrogen is consumed rapidly as soon as flow restriction begins to have an effect, and the cell begins to stabilise after about 100 s of

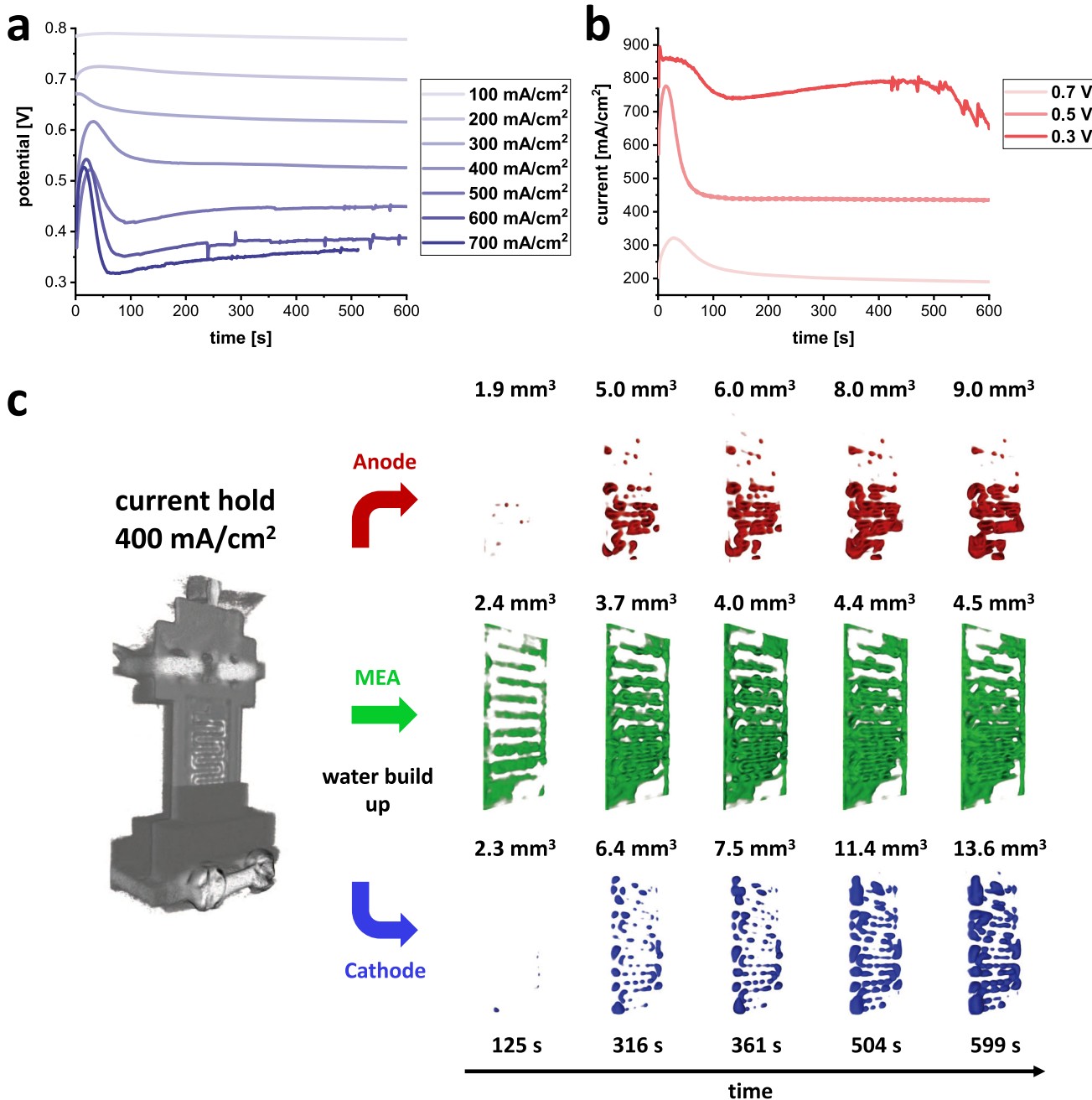

**Fig. 2 Operating parameters and segmented water evolution in the single serpentine anode and cathode flow fields. a** Voltage curves for current holds from 100 mA cm$^{-2}$ to 700 mA cm$^{-2}$; **b** Current curves for potential hold values 0.7, 0.5 and 0.3 V. **c** Water volume build up during cell operation at 400 mA cm$^{-2}$ in the anode and cathode channels. There is less water formation in the anode channel than in the cathode, with a higher concentration in the lower cell section. The cathode displays a more homogeneous water distribution with a slightly increasing water volume gradient towards the bottom of the cell. The MEA exhibits a steady accumulation of water in the first 500 s before stabilising.

operation. Small potential and current peaks and troughs at higher currents are the result of an uncontrolled cell flooding, e.g., at 600 mA cm$^{-2}$ current hold or at 0.3 V potential hold.

**Water evolution in the anode and cathode flow fields.** The generated water in the cathode (blue) and anode (red) single serpentine flow fields, as well as in the MEA (green), are visualised by applying a greyscale threshold to isolate the water in the flow channels, as described in the "Methods" section. Figure 2c shows the 3D rendered water volumes inside both flow fields and the MEA at selected time intervals for the 400 mA cm$^{-2}$ current hold measurement. During operation, water accumulates in both

flow fields whereby the redox reaction of the diffused protons and oxygen from the air occur on the cathode side. Accumulated water on the anode side is the result of back diffusion, where water is transferred through the electrolyte from cathode to anode. At the anode side, water accumulates preferentially in the lower section of the cell, whereas the cathode side shows a more homogenous water build-up over the whole flow field area. Nonetheless, the cathode side does have a slight gradient in the water distribution, with a larger amount accumulated in the lower channel bends of the cell. Accumulation of water in the lower parts of the cell is attributed to both gravitational force draining water down through the channels, as well as the downward-

flowing gas stream carrying water downwards through the cell. In both cases, the gravity and gas flow are evidently sufficient to overcome the adhesion force holding the water on the channel walls, as may be the case when the water droplets are large enough[30]. It is notable that the largest water droplets are found in the channel bends owing to the increased adhesion forces, due to the higher surface areas, which facilitates water condensation[17]. The 3D visualisation of the water accumulation for all operation conditions can be found in Supplementary Fig. 3 to Supplementary Fig. 12. While previous radiography studies have already shown that water in a serpentine flow field accumulates at the bends of the flow channels, this work highlights the separation of water formation in cathode and anode flow fields, and moreover the water management inside the MEA. Furthermore, time-resolved 4D neutron CT opens the way for a quantitative evaluation of the local water evolution, such as the formation, growth and spatially-resolved dynamics of water droplets, and droplet wetting on flow channel surfaces.

Given the 4D nature of this study, the cathode and anode flow channel water contents could be separated and quantified as a function of time. Figure 3a, b shows the volumes of water accumulated over time under the various load conditions at the cathode side. For both conditions (galvano- and potentiostatic holds) a more rapid water volume accumulation is found for the cathode than for the anode (Fig. 3c, d), as expected given the formation of water on the cathode side during cell reaction. There was a continuous increase of the water volumes with time for current holds from 100 to 700 mA cm$^{-2}$, expected due to the higher reaction rate at higher currents. This confirms the findings of earlier research using hydro-electro-thermal measurement techniques[31] or microscale X-ray CT imaging of the GDL[32] for studying water evolution in PEFCs during operation. During the 700 mA cm$^{-2}$ current hold, the volume of water reaches a plateau of about 13 mm$^3$ after around 500 s. This highlights that in these early stages, directly after application of the load, water quantities in the channels approaches, but does not reach, an equilibrium state for which the rate of water production equals the rate of water removal from the flow channels. The curves for the potential holds display a similar behaviour, with a continual increase of water volume up to a maximum of 16 mm$^3$ for the hold voltage of 0.3 V.

Figure 3c, d shows the corresponding curves for water accumulation in the anode flow field. Interestingly, for the galvanostatic holds, water volumes steadily increase with the current density up to 400 mA cm$^{-2}$. For this hold point, the amount of water gradually increases up to a maximum at 480 s, after which part of the water is removed from the cell channels. This behaviour can be seen for the higher hold currents as well, with water volumes levelling off and decreasing. The maximum amount of water in the anode flow field is 4–6 mm$^3$, increasing with current density. The quantified water volume build-up and removal show good agreement with the water amounts visualised in Supplementary Figs. 3–9. The water volume grows and retreats as plotted in Fig. 3. The local distribution and rearrangement of the water droplets is given by the 3D rendered volumes such as given in Supplementary Fig. 7 for the anode flow field at 500 mA cm$^{-2}$, where the water amount is observed to significantly decrease in the time between 475 and 571 s. Figure 3d quantifies the water build-up inside the anode flow field under potentiostatic holds of 0.7, 0.5 and 0.3 V and the results show similar trends as the galvanostatic hold curves, with a greater volume of water being produced for a lower voltage (corresponding to a higher current density). The amount of water produced at 0.5 and 0.3 V display saturation at about 4 and 7 mm$^3$, respectively.

As expected from the half reactions of the cell, the volume of water accumulated in the cathode flow channels is greater than that in the

anode flow channels for all load conditions. For the 700 mA cm$^{-2}$ current hold, maximum water volumes of 13 mm$^3$ for the cathode and about 6 mm$^3$ for the anode are observed. In the case of the potential hold conditions the maximal values are >15 mm$^3$ and about 7 mm$^3$ for the cathode and anode, respectively. However, the presence of water in the anode highlights that there is significant back diffusion to the anode during operation, which results in water accumulation in the flow channels. The ability to study the water accumulation in 3D is a significant benefit, as the nature of water management is complex. The rate of water formation at the cathode is expected to correlate with the electrical current generated due to electrochemical reactions; higher currents will result in more water formation. There is an additional transient aspect to the residence time of water in the channels, observed here, as water obviously accumulates in the channels if the rate of production exceeds the rate of removal. Thus, the decrease in water volume at higher current densities/lower potentials in the anode indicates that the crossover rate is lower than the removal rate and the water volume starts to fall. It should also be noted that there are additional changes of pressure along the channel as a result of water droplet formation. The mobility and coalescence of droplets also alters the average contact area between droplets and flow fields. Thus, the *operando* work presented here provides a significant step in the opportunity for combining experimental and modelling studies of water droplet behaviour in the flow channels. Furthermore, future studies will extend the observation period beyond 600 s (i.e., beyond the "load application" conditions used here), to understand the dynamics of water filling/removal during steady-state operation over longer time periods.

In order to establish a connection between the water evolution and the current and potential, linear fits of the water volumes over the current and potential were produced (Fig. 3a, d), using the linear sections only, i.e. not using the first seconds, to take into account the non-linear behaviour of the cell in the moments directly after application of the load (see over-shoots in Fig. 2a, b) and not using saturated or falling sections. The linear fits yield useful relations between the water volume increase per unit time in connection to the applied current and potential, in mm$^3$ cm$^2$ (A min)$^{-1}$ and mm$^3$ cm$^2$ (V min)$^{-1}$, as given in Fig. 3e, f, respectively. The volume versus potential curves graphs have negative slopes because of the higher currents at lower potentials.

The linear fits in Fig. 3e, f show only a small deviation from the determined values from Fig. 3a, d. Only for low currents <200 mA cm$^{-2}$ does the water content deviate from a linear behaviour. For the linear region, the current-dependent water evolution per unit time results in a water increase of 2.6 (mm$^3$ min$^{-1}$) (A cm$^{-2}$)$^{-1}$ for the cathode and 1.6 (mm$^3$ min$^{-1}$) (A cm$^{-2}$)$^{-1}$ for the anode side. Thus, the total rate of water injection into the flow channels is 4.2 (mm$^3$ min$^{-1}$) (A cm$^{-2}$)$^{-1}$. In the case of the potential hold, values of |−3.6 (mm$^3$ min$^{-1}$) V$^{-1}$| for the cathode side and |−2.7 (mm$^3$ min$^{-1}$) V$^{-1}$| for the anode side are obtained, which yields a rate of water injection into the flow channels of 6.3 (mm$^3$ min$^{-1}$) V$^{-1}$.

**Water evolution in the PEFC membrane electrode assembly (MEA).** The investigation of the water management in the MEA is of great interest for fuel cell improvements, because of the impact of water distribution on the fuel cell performance. For example, poor water transport away from the electrolyte, as well as water back diffusion, can result in unwanted water diffusion towards the anode side and a corresponding drop in fuel cell performance. One of the main limitations of a radiography setup is the inability to differentiate between the water in the flow field channels and the MEA during image analysis, which is overcome by the 4D imaging method as shown here. It should be noted, however, that in this study, the MEA is treated as a single-layer

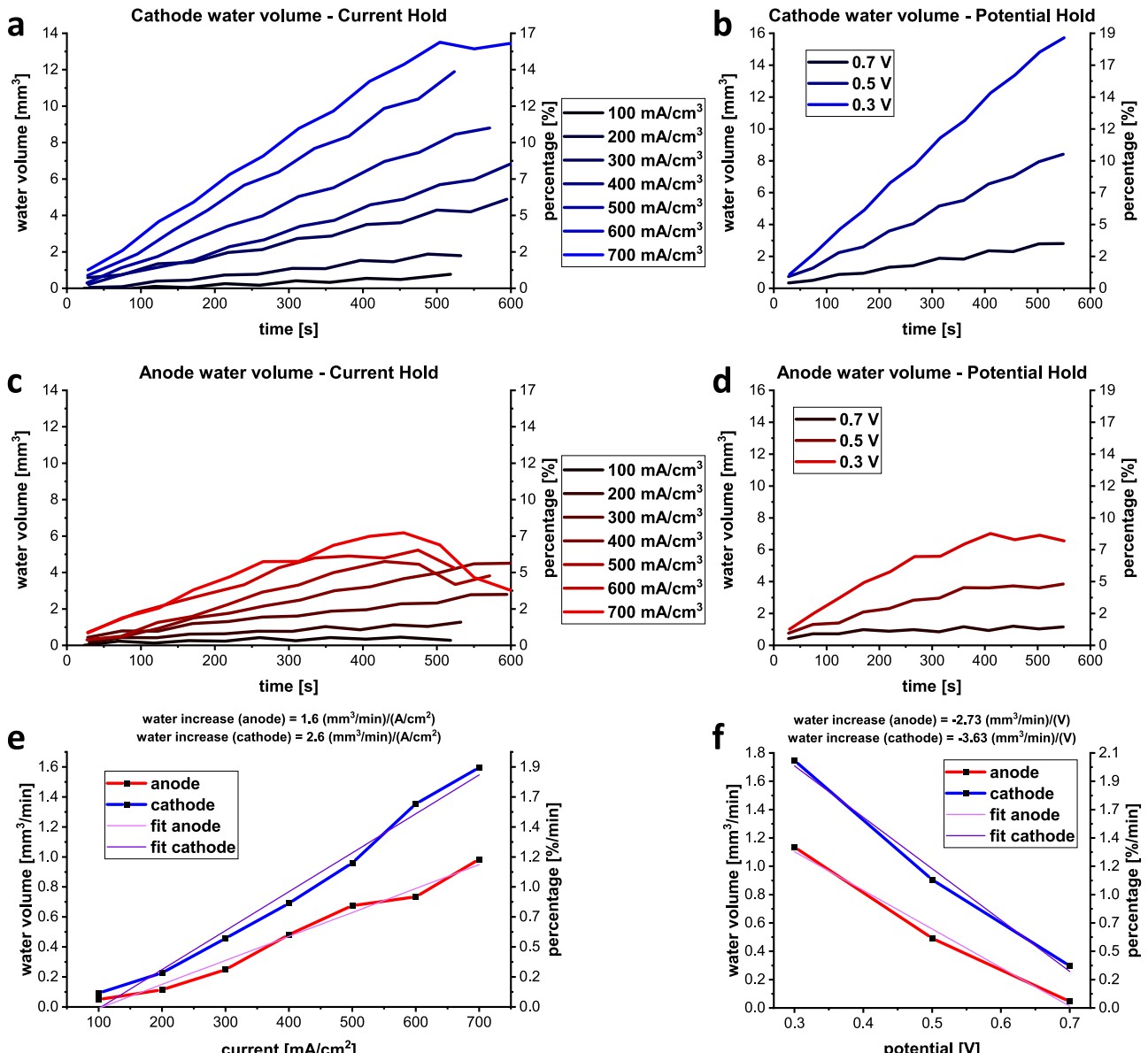

**Fig. 3 Water volumes in the cathode and anode flow fields during constant current and potential hold.** The water volumes increase with increasing current density and decreasing potential for the cathode (**a**, **b**) and anode (**c**, **d**) side with a greater water volume accumulating at the cathode. After a short time, the water amount appears to reach a maximum dependent on the current and potential height before starting to expel water. Graphs **e** and **f** show the water volume changes as a function of the current and potential, respectively, for anode and cathode flow fields. The slopes of the linear regressions represent the current/voltage and time-dependent water volume changes.

component, since the imaging conditions were optimised for temporal rather than spatial resolution.

In the dry state, the MEA is easily distinguished by brighter greyscale values, resulting from higher neutron attenuation due to hydrogen in the Nafion layer and in the platinum catalyst of the CLs, sandwiched between the darker anode and cathode aluminium flow fields. The 3D, threshold segmented images of the MEA (green), shown in Fig. 2c, allow a first visual analysis of the water evolution during the different operational conditions of the PEFC. In general, the MEA fills up with water during operation and the amount of water generally increases depending on the pre-set current density. A higher current or, consequently, a lower voltage causes a higher water production. At lower currents, and during the first phase of cell operation, the MEA shows good expulsion of water into the flow fields. A higher water accumulation is detected behind the flow field ribs, as a result of greater compression of the

pores in the GDL under the flow field rib regions. This compression causes a poorer water evacuation in those areas and consequently a loss of reactivity and efficiency of the MEA and the fuel cell. This has been shown using modelling and experimental studies at the microscale[33–37]. However, the large FoV of the neutron CT provides a comprehensive overview of the water management of the whole MEA area where previous X-ray studies had limited oversight, focusing only on one or two rib/channel interfaces. In terms of the dynamics of water evolution during the current holds, the MEA fills with water continuously, starting behind the flow field ribs, then in the lower cell section, and followed by filling the upper section. For higher current densities, e.g., 700 mA cm$^{-2}$ (see Supplementary Fig. 9), and longer operation times, the water amount begins to reduce, which is caused by the water discharge from the flow field to the cell gas outlets facilitating the water evacuation from the GDL pores in the flow field channels.

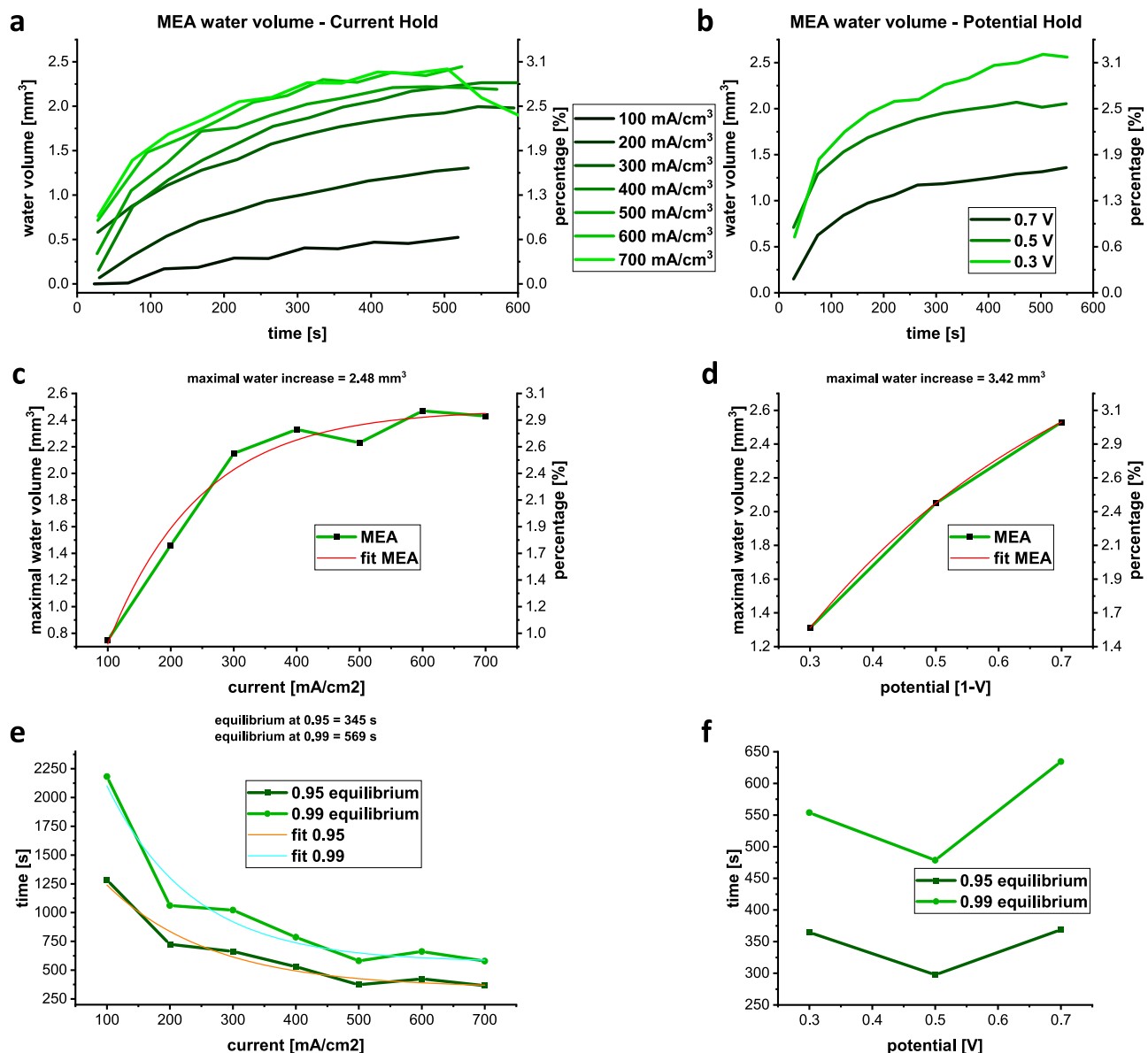

**Fig. 4 MEA water evolution, calculation of the current and potential-dependent water formation and time to reach the water volume equilibrium. a** and **b** The volume increase of the water generated inside the MEA for constant current density and potential, respectively. Higher currents/ lower potentials cause a faster water uptake, which saturates for higher operational cell power. **c** and **d** Determine the maximal water volumes in the MEA. To use the same saturation function the potential values in **d** are reversed with a maximal open circuit potential of 1.0 V. **e** and **f** Evaluates the time needed to reach the water equilibrium for given current and potential densities of 95 % and 99 % of equilibrium, with corresponding fit functions for **e**. A fit for **f** is not possible due to the large uncertainties and small number of measurements.

A quantitative analysis of the water volumes inside the MEA is performed as described in the "Methods" section and as applied for the flow fields. Figure 4a, b displays water volume inside the MEA for the seven current densities (a) and the three potential holds (b). With increasing current (a), and decreasing potential (b), asymptotic behaviour is observed, which arises due to the gradual filling of the MEA up to a volume limit around 2.5 mm³.

The maximum water volumes in the MEA were determined by fitting a saturation function given in Eq. (1).

$$f(t) = a\left(1 - e^{-\frac{t}{\tau}}\right) + b \qquad (1)$$

The determined maximum water volumes show a current density and potential dependency, as depicted in Fig. 4c, d. For increasing currents, the maximum water contents increase up to 400 mA cm⁻². For the dependency with the potential, a similar

characteristic is considered with larger uncertainties due to the small number of measuring points. In the case of the potential hold, the potential values are reversed by the expected maximal potential value of 1.0 V, which represents a typical maximum open circuit potential for the studied fuel cell under real conditions, to enable the use of the saturation function. The maximal water volumes in the MEA under current and potential conditions are 2.5 and 3.4 mm³, respectively.

For the equilibrium state, the emerging amount of water from the redox reaction is equal to the amount that is evacuated into the flow field channels when the MEA has reached its maximum water level. This is the case when the water amounts in the flow fields also reach equilibrium. The time, $t_{equilibrium}$, that is needed to reach the equilibrium state can be calculated by rearranging the saturation function, Eq. (1), as given in Eq. (2). The variable $p$ represents a fraction of the equilibrium and lies between $0 < p < 1$.

Supplementary Table 4 shows the calculated times that are needed to reach 95% and 99% of the equilibrium state of the MEA.

$$t_{\text{equilibrium}} = -\tau \cdot \ln \left( 1 - p \frac{f(t) - b}{a} \right) \qquad (2)$$

For high cell powers, i.e. high current densities and low potentials, a minimum time of operation needed to reach the equilibrium state can be determined. The equilibrium times are plotted over the corresponding currents and potentials in Fig. 4e, f, with fitted decay functions, Eq. (3), where $x$ is the current density or potential, $\tau$ is the tangent in t = 0 s and $a + t_{\text{equilibrium}}$ is the time at 0 mA cm$^{-2}$ or 0 V, respectively:

$$f(x) = a \cdot e^{-\frac{x}{\tau}} + t_{\text{equilibrium}} \qquad (3)$$

For high current densities, the minimum time to reach the equilibrium state in the MEA is 345 s and 569 s, for 95% and 99%, respectively. For the potential hold it was not possible to evaluate these minimum times, given only three points and large uncertainties (Fig. 4f).

**Water evolution in the whole single serpentine fuel cell and comparison with the theoretical water evolution**. To verify the observed amount of water in the PEFC, it is important to understand the possible maximal theoretical generated water amount during cell operation. The ability to calculate water volumes (as opposed to thicknesses in radiography studies) is a significant advantage of this work since quantitative analysis of tomography data allows for a direct comparison with results from cell modelling. The theoretical water production with time depends on the current density and can be calculated as shown in the "Method" section, and compared in Fig. 5a, b with observed data. The theoretical (dashed lines) and the measured volumes (solid lines) show a linear water increase with time. The experimental water volumes increase by about three quarters, as approximately one-quarter of the H$_2$O is expelled from the cell in the gas stream. This is a particular area of interest for future work at higher temperatures, with humidified gases, since the additional heat and moisture is expected to alter the accumulation and expulsion properties of the cell.

Further comparison is possible by measuring the water volume slope of the experimentally determined water volume increase and calculate a current and potential time-dependent water volume evolution, in mm$^3$ cm$^2$ (A min)$^{-1}$ and mm$^3$ min$^{-1}$ V$^{-1}$, for the comparison with the theoretical values.

The water volumes are plotted and linearly fitted over the current density and potential in Fig. 5c, d. For the galvanostatic condition the theoretical slope (5.62 mm$^3$ cm$^2$ (mA min)$^{-1}$) is slightly steeper than the experimental slope (4.45 mm$^3$ cm$^2$ (mA min)$^{-1}$) since part of the produced water is extracted as droplets by the gas flow in the flow field channels. However, the experimental and theoretical curves show a good correlation. This type of analysis provides a direct measurement of the volume of water generated, its movement through the flow channels and removal from the cell, which can vary with multiple factors, such as cell temperature or gas stoichiometry. The ability to quantify the water volume and compare it to theoretical water evolution is a particular advantage of this method. Furthermore, it is expected to be of high value for future practitioners to assess the suitability of novel flow field designs for efficient water removal.

## Discussion

4D time-resolved neutron CT data were obtained from a miniaturised PEFC with a single serpentine anode and cathode flow field

design. The *operando* measurements were performed under various current and potential hold conditions. Due to the high sensitivity of neutrons to the hydrogen, the volume build-up is quantified for both the flow fields and the MEA across the entire cell.

Until now, neutron radiography has been the standard investigation tool to study the water management in PEFCs, but 2D images in- or through-plane only allow for the detection of water path integrated through the particular plane/s. The separation of the produced water in all three cell components is not possible in through-plane measurements and represent the integrals for one component along the neutron path in in-plane mode. Therefore, neutron radiography allows for mainly a qualitative analysis of the water evolution while subtle changes or slight variations of the cell operation may remain undetected. Limited quantitative analysis can be achieved by neutron radiography, such as the calculation of water thicknesses, as average values in the direction of interest. However, the continuous improvements of neutron imaging detectors and installation at high-flux imaging beamlines[38,39] in recent years have enabled 4D measurements of water evolution demonstrated in this work. 4D studies overcome the aforementioned deficits of radiographic neutron imaging and includes all of the benefits such as non-destructivity and a deep penetration through metallic components.

The 4D PEFC measurements provide a framework for fuel cell characterisation where the water build-up can be quantitatively and locally resolved over time under various operational conditions such as constant current or potential holds. Even small changes of the operational conditions can be detected by quantification of the 3D local water build-up. It provides a powerful tool for studying the evolution of water, to focus on individual cell parts such as the cathode and anode flow fields or the MEA, and attention can be paid to details such as water droplets and their dynamic movement along the flow channels. The quantitative analysis of the water transport will lead to a better comparability of different fuel cell designs and configurations based on the quantitative analysis of the water evolution. Fuel cells can be characterised by specific parameters coupled to the water management such as the water accumulation per time in the cathode or anode flow channels or the equilibration times in the MEA. The 4D analysis provides a detailed insight into deficits of fuel cell engineering and may provide a basis for substantial improvements. During the last decade a wealth of modelling work has been done on the evolution of water where the 4D neutron CT presented here can provide realistic validations of models.

This 4D CT technique is not limited to studies of water evolution after load or miniaturised fuel cells. The use of new high-flux neutron imaging instruments such as the NeXT neutron and X-ray tomograph[38] at the Institut Laue-Langevin (ILL, France), will reduce the exposure times per single CT and increase the spatial resolution by scanning PEFCs with larger active area. Moreover, the non-destructive nature of neutrons allows (owing to the lack of radiation damage) long-term measurements in steady-state operation and the study of cell operation under extreme conditions such as high or low temperatures, and with water changing to vapour or ice. Thereby, for each environmental condition the optimal flow field design, e.g. single or double serpentine, parallel or nature-inspired designs, MEA, cell pressure and gas flow can be determined by a straightforward quantitative comparison. Furthermore, while the cell conditions used here (dry gases and room temperature operation) are relevant for portable power applications (like drones or portable chargers), future work should use elevated temperatures and humidified gases to mimic conditions relevant for other, more widely used, PEFC applications, like transport or stationary power.

4D neutron CT provides clear benefits for the characterisation of PEFCs which contain different components, designs and which run

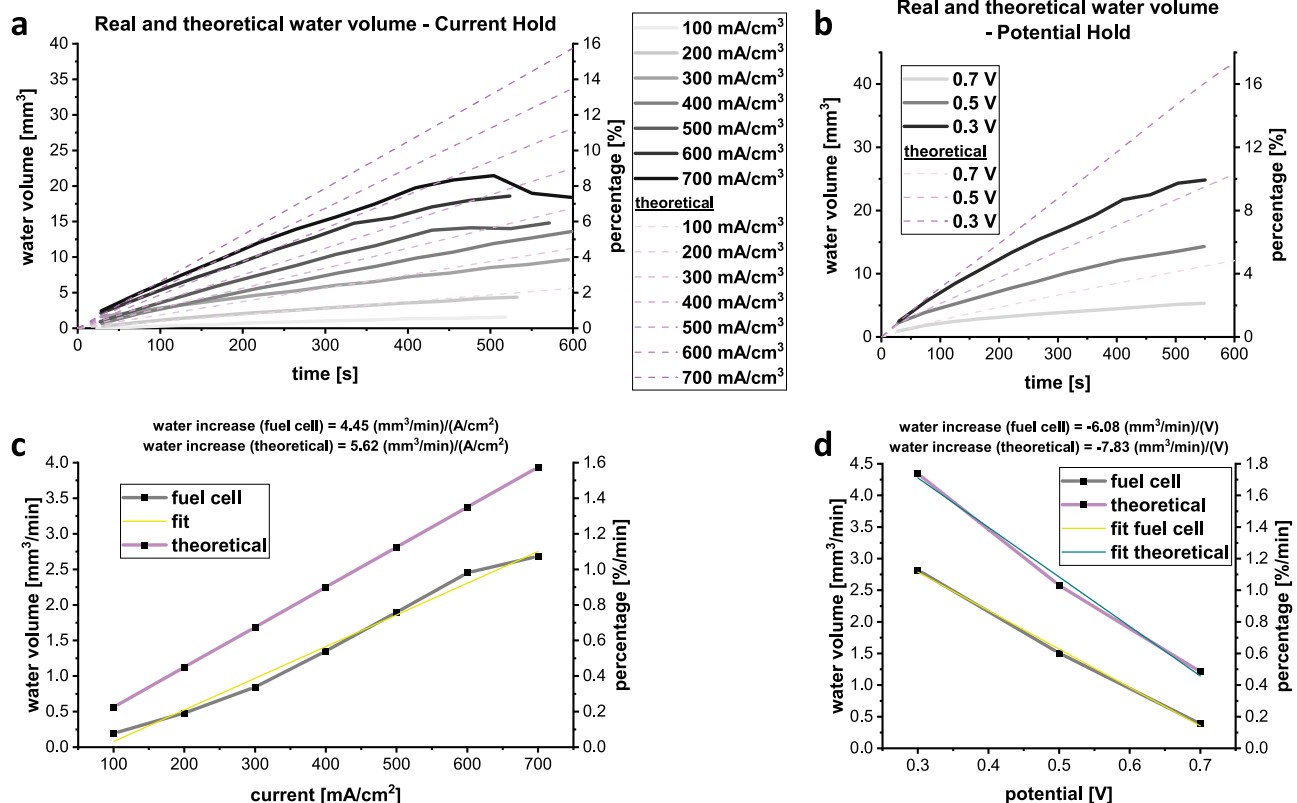

**Fig. 5 Theoretical and observed water evolution inside the single serpentine fuel cell. a** and **b** The theoretical and experimental water volumes for different current densities and potentials as dashed and solid lines, respectively, normalised to an active area of 1 cm². **c** and **d** Compare theoretical and experimentally determined time-dependent water increases over the current and potential.

under different conditions for, e.g. gas flow, cell pressure, and current. Especially the quantification of the water build-up allows a simple way of comparing different cells and of optimising each component. The comparison of the water volume formed in the cell determined from neutron CT with the theoretical volume has been shown in this work. The different sensitivities of neutrons to the hydrogen isotopes can be further exploited to investigate phenomenon such as the back diffusion of water, from the cathode to the anode. Also, for lifetime studies of droplets and water accumulation the use of heavy water can be of particular interest to provide a deeper understanding of the water dynamics and indicate e.g. 'dead' areas providing a high impact for fuel cell engineering [40–42].

## Methods

**Membrane electrode assembly (MEA)**. The 2 cm² MEA was manufactured in-house by hot pressing a membrane of 20 μm thickness (GORE SELECT M8 30.25, W. L. Gore & Associates Inc., USA) to two 250-μm-thick gas diffusion electrodes (HyPlat, South Africa), which utilised Freudenberg H23C9[43] carbon paper. A 12-ton thermal press (Carver, 4122CE) was used to press the assembly with a pressure of 400 psi and a temperature of 150 °C for 3 min. The CLs had a loading of 0.4 mg cm⁻² of platinum on both the anode and cathode. The used hot press conditions are described by Hack et al[7]. The porosity of the GDL in the MEA was about 70 %[44].

**PEFC operation**. The cell compression was about 17 % and the anode and cathode were connected to a GAMRY Interface 5000e potentiostat (Gamry Instruments, USA) for electrochemical control. The cell was operated at ambient temperature, with no additional cell or gas heating. Dry hydrogen and dry air were provided at the lower and upper pipe inlets using flexible pipes on the anode and cathode sides, respectively, with no additional humidification. Gas lines were guided through the 1 mm diameter channels of the flow fields, with flowrates of 20 ml min⁻¹ on the anode and 100 ml min⁻¹ on the cathode. The anode and cathode single serpentine flow field design was mirror-mounted. The unreacted hydrogen, air and the formed water were removed by the lower pipe outlets and guided through the beamline extraction system. A LabView code was used to control the gas flow and the cell parameters. The polarisation curve and power slope of the single serpentine PEFC

were determined once the cell had been mounted onto the beam, to establish the experimental conditions for the galvanostatic/potentiostatic holds. The V-I characteristic was measured from 0 to 0.3 V and plotted in Supplementary Fig. 1a.

**Neutron imaging**. CONRAD-2 at BER II (Berlin) provided cold neutrons from a hydrogen moderator (<30 K) via a curved neutron guide. At 1.4 m downstream the end of the neutron guide the PEFC was mounted on a rotation stage about 50 mm in front of the neutron camera, providing a beam divergence given by L/D~70. Radiographic projections were recorded by an indirect neutron imaging camera with a 400 μm ⁶LiF/ZnS:Ag scintillator screen (Applied Scintillation technologies, UK) and a CMOS ASI178MM-Cool camera (https://astronomy-imaging-camera.com) with a Nikon photo lens (Nikkor 20 mm, Nikon, Japan) with 20 mm focus. A high neutron flux on the imaging position allowed for an exposure time of 100 ms per projection providing a spatial resolution of <300 μm. For a faster readout of the camera and a higher count rate per pixel the 3096 × 2080 pixel CMOS array was binned by 2 resulting in a pixel size of 63.6 μm. The FoV of the camera was 100 × 65 mm², of which 2/3 was illuminated by neutrons restricted by a 30 mm diameter collimator at the end of the neutron guide.

*Operando* CT scans were initiated with the start of the fuel cell operation at the above-mentioned constant current and potential hold values. One CT was collected in about 40 s over a rotation of 2π with 394 projections per tomogram and an exposure time of 100 ms per projection. Between each tomogram acquisition, (i.e. rotation 370 ° forwards and 370° backwards), the beamline shutter was closed and opened for a better differentiation between successive CT data sets. The measured angular range was adopted to compensate errors from the synchronisation between shutter closures and cell rotations. Figure 1c shows a rendered 40 s 3D volume and the resulting sinogram for the 400 mA cm⁻² current hold. The black vertical stripes in the sinogram mark the rotation reversal times as a result of shutter closures. Between each galvanostatic/potentiostatic hold, the neutron shutter was closed during cell purging, to limit the radiation exposure of the cell, thus minimising any potential beam damage caused by the neutron beam. It was verified that there had been no beam damage, by comparing the average current of the final experiment (0.3 V potential hold), of 772 mA cm⁻² with the polarisation performance at the beginning of the test (0.34 V at 700 mA cm⁻²) and the average performance during the 700 mA cm⁻² galvanostatic hold (0.36 V).

**Tomographic reconstruction and water volume quantification**. The projections were dark-field corrected and normalised by the open beam. Sinograms were stripe

and noise filtered by using a 1-dimensional (1D) orientated median filter. All backward-rotation projections were arranged in reverse order, to prevent mirrored volume data in the time-sequence of tomograms. Due to the unsynchronised image recording and fuel cell rotation, a data sorting step was necessary to find the first and last projection of each 360 ° subset, in such a way that the time-separated 3D reconstructed volumes were aligned. This was achieved manually, starting with the 100 mA cm$^{-2}$ scan, by producing ratios and averages of projections for each sinogram in a range of angles, and by selecting the first and last projection for a 0–360° for which the averaged ratio was close to unity. The number of projections for a 360° reconstruction subset was found to be 394. The centre of rotation (CoR) and rotation axis tilt were determined manually for the first tomogram of the 100 mA cm$^{-2}$ scan by determining CoR for an upper and lower cell section. For the 3D reconstruction the algorithm SIRT (simultaneous iterative reconstruction technique) with 400 iterations as implemented in the ASTRA toolbox[45,46] was used within the Python programming language. SIRT was used due to its superiority over filtered back projection for very noisy projections. In the post-reconstruction, the data sets were horizontally and vertically aligned in all three directions.

For the 3D visualisation of the time-dependent water build up a threshold segmentation was used, as shown in Fig. 2c. But due to the large voxel size a more accurately sub-pixel method was utilised for the quantification and for comparison of the water volumes. In a first step, the related water volumes in the anode and cathode flow fields and the MEA were located and defined in the tomographies. The thicknesses of the anode and cathode regions in the imaging data were found to be about 1.1 mm, slightly larger than the channel thickness, owing to the coarser spatial resolution of the tomography setup. The thickness of the MEA was determined to be about 600 μm, thus matching the design value. Owing to the large voxel size of 63.6 μm a thresholding segmentation approach yielded a large error for the water volumes. More precise results were obtained by determining the relative water volume increases, as percentage changes with regard to the dry fuel cell. Equation (4) was used for the water volume estimation, $V_{H_2O}$, in a volume $V$, e.g. anode flow field volume, in a certain tomogram $i$. $Int_i$ and $Int_{dry}$ represent the average grey value intensities of the volumes $V$ in tomogram $i$ and a reference tomogram of the dry fuel cell. $Int_{H_2O}$ is the intensity of 100% water.

$$V_{H_2O} = \left(\frac{Int_i - Int_{dry}}{Int_{H_2O}}\right)V \quad (4)$$

**Calculation of the theoretical water evolution**. The volume of generated water is linearly dependent on the current density. The anode and cathode-based catalytic reactions are shown in Eqs. (5) and (6). Equation (7) combines both reactions in one overall reaction for the water generation in a PEFC. For the generated amount of water, the number of transported hydrogen ions H$^+$ (protons) and consequently the number of electrons e$^-$ are important. Per reacted water molecule, two hydrogen ions are needed which results in a produced charge of 2·1.6022·10$^{-19}$ C per molecule, see Eq. (7). Following on, a theoretical water production density $W_{H_2O\ theor}$ is derived, i.e., the produced water volume per time and cell area. The relationship with the current density is given by Eqs. (8) and (9)

$$\textbf{Anode } 2\,H_2 \rightarrow 4\,H^+ + 4\,e^- \quad (5)$$

$$\textbf{Cathode } O_2 + 4\,H^+ + 4\,e^- \rightarrow 2H_2O \quad (6)$$

$$\textbf{Overall reaction } 2H^+ + 2e^- + 0.5O_2 \rightarrow H_2O \quad (7)$$

$$W_{H_2O\,theor}\left[\frac{cm^3}{s \cdot cm^2}\right] = \frac{\eta_{H_2O}\left[\frac{mol}{s \cdot cm^2}\right] \cdot M_{H_2O}\left[\frac{g}{mol}\right]}{\varrho_{H_2O}\left[\frac{g}{cm^3}\right]} \quad (8)$$

$$\eta_{H_2O}\left[\frac{mol}{s \cdot cm^2}\right] = \frac{j\left[\frac{A}{cm^2}\right]}{2e[As] \cdot N_A\left[\frac{1}{mol}\right]} \quad (9)$$

where $M_{H_2O}$ is the molecular mass of water, $\rho_{H_2O}$ is the water density and e the electron charge. $\eta_{H_2O}$ represents the produced substance amount density and $N_A$ is the Avogadro constant. The above relations permit calculating a theoretical water volume which is produced from a current density of 1 A cm$^{-2}$ in one second, which is 937.5·10$^{-4}$ mm$^3$ [47].

## Data availability
The data supporting this study are available from the corresponding author upon request.

## Code availability
The code supporting this study is available from the corresponding author upon request.

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

## Acknowledgements
This work was carried out with funding from The Faraday Institution (faraday.ac.uk; EP/S003053/1), grant number FIRG0013 (Characterisation Project). The authors would like to acknowledge the Royal Academy of Engineering (CiET1718\59) for financial support. Jennifer Hack acknowledges a studentship from the EPSRC CDT in Advanced Characterisation of Materials (EP/L015277/1) and an EPSRC Doctoral Prize Fellowship (EP/T517793/1).

## Author contributions
R.F.Z., J.H and P.R.S. conceived the study. J.H. designed and optimised the fuel cell for neutron imaging. R.F.Z. performed the 3D data reconstruction and analysis of the water volume. R.F.Z. and J.H. led the experimental work and analysis. N.K. and H.M., beamline scientists on the CONRAD neutron imaging beamline at HZB, supported the neutron imaging of the fuel cell. L.R., M.M., C.T. and T.M.M.H. helped to run the experiments with their expertise. I.M., W.K., D.J.L.B., and P.R.S. supported the work with their expertise in fuel cell science and neutron imaging and helped to design the experiment and analysis. All authors have contributed to writing and reviewing the manuscript.

## Competing interests
The authors declare no competing interests.
