## [Peer Review File · Nature Communications]

High-Speed 4D Neutron Computed Tomography for Quantifying Water Dynamics in Polymer Electrolyte Fuel CellsREVIEWER COMMENTS

Reviewer #1 (Remarks to the Author):

A new development of a 4D neutron imaging technique was built through this study for an essential problem which is the water management inside the operating Polymer Electrolyte Fuel Cells PEMFCs. Using this model, the authors verifying a quantitative analysis of the local water evolution dynamically.

The problem of water management in PEMFCs is one of the most important problems facing the use of these cells in various engineering applications. The increase in the presence of water in the cell leads to a blockage in the reactant gas supply channels and consequently poor performance. On the other hand, the decrease in the presence of water inside the cells leads to dehydration of the cells and thus affecting the Protons conductivity. Hence, the study of dynamic water management within the PEMFCs is the most important study. Through this research, the researchers succeeded well in determining the amount of water produced and its distribution mechanism using neutron imaging. Thus, this study is one of the important studies to develop the work of this type of fuel cell.

The topic of the quantifying water dynamics inside the Polymer Electrolyte Fuel Cells PEMFCs has significant importance and the paper "High-Speed 4D Neutron Computed Tomography for Quantifying Water Dynamics in Polymer Electrolyte Fuel Cells" is of potential interest for publication as the studying of the operating PEMFCs as well the dynamic behaviors of the water during the operation of the PEMFCs are an advance fundamental issue which will improve the using of the PEMFCs in the field of engineering and technology. However, some minor points need to be improved.

- 1- The English around all the paper needs to be checked.
- 2- It would be better if a separate section about the experiment was added independently from the results, as the mixing between the experiment works with the results leads to confusion.
- 3- The first figure contains a lot of information that confuses the reader, so it is recommended to clearly identify the sub-images
- 4- In figure 2.a there are some clear peaks when using the current of 600mA/cm² between the time started from the 250s to 600s please explain the reason for this.
- 5- There is no need to mention some general equations such as equations 5 till 9

Finally, I recommend publishing this work as it will be a good added information for researchers all over the world.

Reviewer #2 (Remarks to the Author):

Title: High-Speed 4D Neutron Computed Tomography for Quantifying Water Dynamics in Polymer Electrolyte Fuel Cells

This manuscript presents a study on the visualization of liquid water, in an operating PEFC. Key results include the dependency of liquid water amount on operating current density in the flow channels and MEA. This is demonstrated by electrochemical experiments and imaging combined with analytical calculations for validation. The use of the neutron imaging technique at increased time resolution with associated analysis is a key contribution of this work – presenting more experimental understanding of temporal water distribution over a large domain for varying operational current densities. However, the manuscript may be improved in the following areas:

1. Since a key part of this work include water distribution dependence on current density, the authors should discuss the results in this work as it relates to some existing literature in this area. There are existing works on the influence of current density on water distribution, albeit using different method – it is worth pointing out how this work compares to those.

2. Importantly, what new knowledge is generated compared to previously published X-ray CT reports which show water in GDLs and flow fields of similar cells? Compare for example, to the work of Eller et al. (PSI) using X-ray CT with sub second temporal resolution, showing water in both GDLs and flow fields. 4D water imaging in a fuel cell is not new.
3. The operating conditions of the cell, which are key water distribution have not been clearly stated. Conditions such as humidity and temperature as well as typical hydrogen/oxygen stoichiometries should be stated to adequately interpret the results. Note that if dry gases and room temperature were used, this must be clearly mentioned within the manuscript because such conditions are very unusual and do not represent normal fuel cell operation, and if so, the results shown are not representative of regular fuel cell operation.
4. How does the water volumes in figure 2 compare to the formula in equation 4? It appears similar inferences are made from both methods, but the authors have pointed that one was used for volume estimation. More details on this should be provided in the section on water volume quantification as a major part of the results presented depend on this.
5. Time scale of water transport: it is important to state the capabilities and limitations of the present technique to capture water droplets moving rapidly through the GDLs and flow fields. It is well known that such dynamics occur on the order of sub seconds to seconds, whereas the temporal resolution of the present technique is reported to be 40 s. Are the results shown averaged over a 40 s period? If so, what are the implications of this on the water volume results in a dynamic system?
6. Did the authors verify that there was no irradiation damage to the MEA or system caused by the neutron beam? Given the higher flux than in previous studies, such verification is important and must be reported.
7. Corrections required: fuel cell membrane made of "PTFE"; "half reaction" (introduction)

REVIEWER COMMENTS

Comment to editor:

we would like to thank the reviewers for their very constructive reports which have helped us to improve the quality of our manuscript considerably. We are pleased to note that the referees do recognise that our 4D neutron imaging study is important for understanding of the water management of PEFCs. We understand the concerns of Reviewer 2, but hope that our responses sufficiently highlight that the concerns do not compromise the study in any substantial way. Please find our point-by-point responses below. We have highlighted the changes in the revised version of the manuscript as tracked-changes. We hope that our manuscript can now be accepted as a paper for Nature Communication.

Reviewer #1 (Remarks to the Author):

A new development of a 4D neutron imaging technique was built through this study for an essential problem which is the water management inside the operating Polymer Electrolyte Fuel Cells PEMFCs. Using this model, the authors verifying a quantitative analysis of the local water evolution dynamically.

The problem of water management in PEMFCs is one of the most important problems facing the use of these cells in various engineering applications. The increase in the presence of water in the cell leads to a blockage in the reactant gas supply channels and consequently poor performance. On the other hand, the decrease in the presence of water inside the cells leads to dehydration of the cells and thus affecting the Protons conductivity. Hence, the study of dynamic water management within the PEMFCs is the most important study. Through this research, the researchers succeeded well in determining the amount of water produced and its distribution mechanism using neutron imaging. Thus, this study is one of the important studies to develop the work of this type of fuel cell.

The topic of the quantifying water dynamics inside the Polymer Electrolyte Fuel Cells PEMFCs has significant importance and the paper "High-Speed 4D Neutron Computed Tomography for Quantifying Water Dynamics in Polymer Electrolyte Fuel

Cells” is of potential interest for publication as the studying of the operating PEMFCs as well the dynamic behaviors of the water during the operation of the PEMFCs are an advance fundamental issue which will improve the using of the PEMFCs in the field of engineering and technology. However, some minor points need to be improved.

We thank Reviewer #1 for their time on this manuscript and hope to address all concerns sufficiently.

1- The English around all the paper needs to be checked.

Thank you for the comment. We have thoroughly checked the English in the paper and made corrections (marked in red), such as:

Line 124; page 4: “A range of galvanostatic and potentiostatic hold experiments were carried out, whilst constantly collecting 40-second tomograms.”

Line 157; page 7: “and oxygen from the air occur at on the cathode MEA side”

Line 212; page 8: “**Error! Reference source not found.** d quantifies the water build-up inside the anode flow field under potentiostatic holds of 0.7 V, 0.5 V and 0.3 V and the results show similar trends as the galvanostatic hold curves, with a greater volume of water being produced for a lower voltage (corresponding to a higher current density). The amount of water produced at 0.5 V and 0.3 V display saturation at about 4 mm³ and 7 mm³, respectively.

2- It would be better if a separate section about the experiment was added independently from the results, as the mixing between the experiment works with the results leads to confusion.

We thank the reviewer for that helpful comment but refer to the formatting instructions from Nature Communication (<https://www.nature.com/documents/ncomms-formatting-instructions.pdf>) which does not permit a separate experimental section. Therefore, we implemented at the beginning of the Results section a short experimental overview comprising the first 4 paragraphs and Figure 1 and 2. A more detailed description of the PEFC used and the experimental details can be found in the Method section as mentioned in the text. Furthermore, we would like to point to

the word limit of 6,000 words, which precludes a more detailed description of the experimental parameters in the main text.

3- The first figure contains a lot of information that confuses the reader, so it is recommended to clearly identify the sub-images

Thanks for this helpful comment. We agree that Figure 1 contains a lot of information, which supports the understanding of the cell design, mounting, experimental procedure and data reconstruction. To differentiate the sections more clearly we have added larger spaces between the single images and have added additional explanatory text in the figure caption to c in line 133 and d in line 134 as marked in red below:

Figure 1 Single serpentine flow field design, fuel cell and imaging processing. **a** shows the flow field design incorporated in the cell endplates for both anode and cathode with hydrogen and air in- and outlets, respectively. **b** cell assembly on the rotation stage using nylon screws and nuts. **c** Tomograms were collected in 40 s during each of $\pm 370^\circ$ forward and backward rotations in the neutron beam with an L/D collimation of ca. 70. **d** Data processing, including flat fielding projections, generation of time dependent sinograms (below) with water volume build up (increasingly dark grey values from 0 to 12), 3D reconstruction, followed by water quantification in the anode and cathode flow fields, and in the MEA.

Images a and b are more clearly separated and described. With the additional explanations in c and d the reader is more easily guided to the conditions of data

collection (c) and image processing/analysis (d) especially with the sinogram description (below).

4- In figure 2.a there are some clear peaks when using the current of 600mA/cm² between the time started from the 250s to 600s please explain the reason for this.

Thank you for the comment. These peaks are due to the forward and back rotation of the cell causing an unstable current in the cell. Future work will use slip rings to avoid any interruption to the current during the rotation of the cell.

5- There is no need to mention some general equations such as equations 5 till 9.

We do agree that equations 5 to 9 are general and well known to everybody working in this field. However, we believe that they are necessary for the non-specialist and are also needed to show how the theoretical water volume is calculated which is a key strength of this work. We therefore, on this occasion, would like to retain this background theory and is fitting for the broad readership of Nature Comms.

Finally, I recommend publishing this work as it will be a good added information for researchers all over the world.

We thank Reviewer #1 again for their final encouraging comments and for useful recommendations which clearly improved the clarity and quality of the article.

Reviewer #2 (Remarks to the Author):

Title: High-Speed 4D Neutron Computed Tomography for Quantifying Water Dynamics in Polymer Electrolyte Fuel Cells

This manuscript presents a study on the visualization of liquid water, in an operating PEFC. Key results include the dependency of liquid water amount on operating current density in the flow channels and MEA. This is demonstrated by electrochemical experiments and imaging combined with analytical calculations for validation. The use of the neutron imaging technique at increased time resolution with associated analysis is a key contribution of this work – presenting more experimental understanding of

temporal water distribution over a large domain for varying operational current densities. However, the manuscript may be improved in the following areas:

1. Since a key part of this work include water distribution dependence on current density, the authors should discuss the results in this work as it relates to some existing literature in this area. There are existing works on the influence of current density on water distribution, albeit using different method – it is worth pointing out how this work compares to those.

We thank the reviewer for their suggestions and have added the following to highlight that other work has shown the current-water relation, albeit using 2D mapping/ neutron radiography techniques (as in the work by Meyer et al.) and microscale X-ray CT study of a single rib/channel region (as in the work by Xu et al.).

We have added the following in Line 182, page 7:

“There was a continuous increase of the water volumes with time for current holds from 100 mA cm^{-2} to 700 mA cm^{-2} , as **expected due to the higher reaction rate occurring at higher currents. This confirms the findings of earlier research using hydro-electro-thermal measurement techniques ¹ or microscale X-ray CT imaging of the GDL ² for studying water evolution in PEFCs during operation**”

2. Importantly, what new knowledge is generated compared to previously published X-ray CT reports which show water in GDLs and flow fields of similar cells? Compare for example, to the work of Eller et al. (PSI) using X-ray CT with sub second temporal resolution, showing water in both GDLs and flow fields. 4D water imaging in a fuel cell is not new.

We thank the reviewer for their acknowledgement of the existing 4D studies of water using X-ray CT. Whilst we agree that 4D water imaging has been done with X-rays, our work presented here is, to our best knowledge, the first 4D water imaging study using neutrons. This is the key novelty presented here. The new information gained is the ability to resolve water across the entire anode and cathode flow fields during operation. Thus, we have been able to directly quantify the total volume of water in the cell and relate it to the theoretical water production, which has not been shown before.

Whilst X-ray CT has advantages over neutron imaging with respect to spatial resolution, the field of view in X-ray CT is limited, which limits the study of global water evolution across the entire cell; moreover there is lower sensitivity to water and a risk of beam damage as discussed below. Thus, we strongly believe that our neutron CT work shown here is complementary to X-ray CT imaging methods for understanding water dynamics in fuel cells.

We acknowledge that the excellent work by Eller et al. using X-rays has shown water in the flow fields, but their work only looked at water in a small section of the cell. In contrast, our 4D neutron technique allows for the study of water in the entire flow field, which we expect to be of particular importance in the future as novel flow field designs evolve. The application of our 4D method can also be applied to larger cells and fuel cell stacks, which would not be possible with X-ray techniques because of the limited field of view. Furthermore, the contrast between water and the fuel cell components is much poorer using X-ray techniques compared with neutron techniques, which allows for an easier quantification of water in the flow channel.

Additionally, in relation to the reviewer's comment #6 below, we would also like to mention that the intensity of X-ray beams used for high spatially resolved X-ray CT is several magnitudes higher than the flux of neutron studies. Resulting interaction between X-rays and e. g. the proton exchange membrane can destroy molecular bonds or cause cell heating leading to decreasing performance and membrane damage during minutes down to seconds, which is rarely mentioned in most related X-ray studies. We like to point to the work from Eller et al. ³ and Roth et al. ⁴ which describe in detail degradation effects due to synchrotron irradiation e. g. in the GDL and PTFE foil during seconds of X-ray exposure. Thus, neutron studies have a clear advantage, since they do not suffer from the same beam damage effects as studies using X-ray irradiation.

3. The operating conditions of the cell, which are key water distribution have not been clearly stated. Conditions such as humidity and temperature as well as typical hydrogen/oxygen stoichiometries should be stated to adequately interpret the results. Note that if dry gases and room temperature were used, this must be clearly mentioned within the manuscript because such conditions are very unusual and do not represent

normal fuel cell operation, and if so, the results shown are not representative of regular fuel cell operation.

We agree that fuel cell operating conditions are very important. While the gas conditions were mentioned in the “PEFC operation” section of the Methods, we realise we did not state the ambient temperature operation and thank the reviewer for pointing this out. We have added the following to make these conditions clearer.

Line 477; page 17: “The cell was operated at ambient temperature, with no additional cell or gas heating. Dry hydrogen and dry air were provided at the lower and upper pipe inlets using flexible pipes on the anode and cathode side, respectively, with no additional humidification.”

Whilst room temperature and dry gases are not the common operating conditions, especially for automotive application, when considering applications like portable power (phone chargers or drones, for example), the fuel cells in these cases are room temperature with non-humidified gases.

Finally, we acknowledged in our ‘future work’ section the need to carry out further studies at higher temperatures with humidified gases. As the reviewer mentions, these can change the water distribution in the cell and this will provide a good opportunity to better understand the water evolution under these conditions, using our 4D method described here. We have added extra emphasis of the need for such studies as follow on work:

Line 381; page 14: “The experimental water volumes increase by about three quarters, as approximately one quarter of the H₂O is expelled from the cell in the gas stream. This is a particular area of interest for future work at higher temperatures, with humidified gases, since the additional heat and moisture is expected to alter the accumulation and expulsion properties of the cell.”

Line 452; page 16: “Furthermore, while the cell conditions used here (dry gases and room temperature operation) are relevant for portable power applications (like drones or portable chargers), future work should use elevated temperatures and humidified gases to mimic conditions relevant for other, more widely used, PEFC applications like transport or stationary power.”

4. How does the water volumes in figure 2 compare to the formula in equation 4? It appears similar inferences are made from both methods, but the authors have pointed that one was used for volume estimation. More details on this should be provided in the section on water volume quantification as a major part of the results presented depend on this.

We thank the reviewer for this suggestion. The accurate differentiation between the visualisation and the sub-pixel water volume determination is highly important. To be clearer about this, we have rewritten the first part of the water volume quantification section, as follows.

Line 546; page 19: “For the 3D visualisation of the time dependent water build up a threshold segmentation was used, as shown in **Error! Reference source not found.** **c.** But due to the large voxel size a more accurate sub-pixel method was utilised for the quantification and for comparison of the water volumes. In a first step the related water volumes in the anode and cathode flow fields and the MEA were located and defined ...”

5. Time scale of water transport: it is important to state the capabilities and limitations of the present technique to capture water droplets moving rapidly through the GDLs and flow fields. It is well known that such dynamics occur on the order of sub seconds to seconds, whereas the temporal resolution of the present technique is reported to be 40 s. Are the results shown averaged over a 40 s period? If so, what are the implications of this on the water volume results in a dynamic system?

We thank the reviewer for their comments on this and agree that there are temporal limitations due to the current capabilities. Yes, the results show the averaged information of the water build up over a period of 40 s. This makes the technique insensitive for small periodic volume variations <40 s. Bigger volume variations, which are built up or removed during the exposure time, are clearly visible. In this study we analyse the first 10 minutes of the cell start up where small variations of water evolution/removal are unlikely, due to the very linear increase of the quantified water volume under the given conditions. A better temporal resolution is necessary if an equilibrium state between built up water and water removal will be reached in a later phase of operation. At this point we like to refer that the here presented study shows the first results of a non-optimised technique. In future experiments we will increase

the spatial and temporal resolution to $<100\ \mu\text{m}$ and $<10\ \text{s}$ which is comparable to the conditions for most of the shown radiography studies.

6. Did the authors verify that there was no irradiation damage to the MEA or system caused by the neutron beam? Given the higher flux than in previous studies, such verification is important and must be reported.

We thank the reviewer for their concern about irradiation damage and acknowledge that it is a valid concern. However, compared with X-rays, neutrons are inherently more penetrating, with significantly lower flux $<10^7\ \text{neutron s}^{-1}\ \text{cm}^{-2}$ compared to $>10^{13}$ photons $\text{s}^{-1}\ \text{cm}^{-2}$ as mentioned in the last paragraph of question #2. The increase of neutron flux compared to previous studies is not substantial (i.e. not several magnitudes) and only given by the decrease of the distance from the pinhole collimator to the sample from ca. 6 m to 1.4 m resulting in a neutron flux increase <20 times. Further, the interaction probability of neutrons with the sample is much lower than for X-rays (hence the higher penetration), due to the interaction with the nucleus and not with the electron shell resulting in less damage. Neutrons are known to cause irradiation effects on beamlines, e.g. in scintillator materials which however are tuned to efficiently capture neutrons. Neutrons can also deposit heat, however this is observable at sub-Kelvin temperatures only. The increase of measured statistics in terms of signal-to-noise is due to an optimised camera system with high light output and low background as well as that the required dose for a good image is the sum of all projections for a single CT, thus a lack of information in a single projection is overcome by the number of projections. Further, we would like to refer to the study from Matsushima et al. ⁵ which have seen no evidence of beam damage in biological systems, i.e. plants, during long neutron exposure.

To verify that there was no performance loss, we compared the average values for the highest current/lowest voltage holds with the initial polarisation curve. The average voltage during the $700\ \text{mA cm}^{-2}$ hold was $0.36\ \text{V}$, compared with $0.34\ \text{V}$ at $700\ \text{mA cm}^{-2}$ in the polarisation curve, see **Figure 2**. The average current during the $0.3\ \text{V}$ hold (which was the final experiment performed) was found to be $772\ \text{mA cm}^{-2}$. This highlights that there had been no performance loss after the 1.67 h total exposure time during the series of experiments.

Figure 2: Polarisation and power curve of the single serpentine PEFC measured on the beamline.

Finally, the shutter was closed between measurements, while the cell was purged, to limit the neutron exposure to the MEA. Thus, the total irradiation time of the cell was 1hr40 mins, which was not considered sufficient to cause beam damage to the cell.

We have added the following clarification to the text to highlight the steps taken to limit beam exposure, as well as the verification of the lack of beam damage:

Line 518; page 18: “Between each galvanostatic/potentiostatic hold, the neutron shutter was closed during cell purging, to limit the radiation exposure of the cell, thus minimising any potential beam damage caused by the neutron beam. It was verified that there had been no beam damage, by comparing the average current of the final experiment (0.3 V potential hold) of 772 mA cm⁻² with the polarisation performance at the beginning of the test (0.34 V at 700 mA cm⁻²) and the average performance during the 700 mA cm⁻² galvanostatic hold (0.36 V).”

7. Corrections required: fuel cell membrane made of “PTFE”; “half reaction” (introduction)

Thank you to the reviewer for pointing this out and we have updated this in the text for clarity as follows:

Line 37: “PEFCs ~~often~~ typically employ a ~~polytetrafluoroethylene (PTFE)~~ per-fluorinated sulfonic acid (PFSA) membrane...”

Line 42: “Hydrogen is supplied to the anode where the oxidation ~~electrochemical half~~ reaction splits the hydrogen into protons and electrons. The protons are conducted through the ~~PTFE~~ membrane to the cathode where the reduction ~~second-half~~ reaction with oxygen forms water.”

References

1. Meyer, Q. *et al.* The Hydro-electro-thermal Performance of Air-cooled, Open-cathode Polymer Electrolyte Fuel Cells: Combined Localised Current Density, Temperature and Water Mapping. *Electrochim. Acta* **180**, 307–315 (2015).
2. Xu, H. *et al.* Effects of Gas Diffusion Layer Substrates on PEFC Water Management: Part I. Operando Liquid Water Saturation and Gas Diffusion Properties. *J. Electrochem. Soc.* **168**, 074505 (2021).
3. Eller, J. *et al.* Implications of polymer electrolyte fuel cell exposure to synchrotron radiation on gas diffusion layer water distribution. *J. Power Sources* **245**, 796–800 (2014).
4. Roth, J., Eller, J. & Büchi, F. N. Effects of Synchrotron Radiation on Fuel Cell Materials. *J. Electrochem. Soc.* **159**, F449–F455 (2012).
5. Matsushima, U. *et al.* Visualization of water usage and photosynthetic activity of street trees exposed to 2 ppm of SO₂-A combined evaluation by cold neutron and chlorophyll fluorescence imaging. *Nucl. Instruments Methods Phys. Res. Sect. A Accel. Spectrometers, Detect. Assoc. Equip.* **605**, 185–187 (2009).

REVIEWERS' COMMENTS

Reviewer #2 (Remarks to the Author):

The authors have appropriately addressed the reviewer comments. The manuscript can be published.

REVIEWERS' COMMENTS

Reviewer #2 (Remarks to the Author):

The authors have appropriately addressed the reviewer comments. The manuscript can be published.

We would like to thank the reviewers for their very constructive reports which have helped us to improve the quality of our manuscript considerably. We are pleased to note that the referees and the editor accepted our article for publishing.